# Dynamics of Correlated Double-Ionization of Two-Electron Quantum Dots in Laser Fields

**DOI:** 10.3390/ma16041405

**Published:** 2023-02-07

**Authors:** Adam Prior, Lampros A. A. Nikolopoulos

**Affiliations:** School of Physical Sciences, Dublin City University, D09 V209 Dublin, Ireland

**Keywords:** ab initio, double-ionization, energy distributions, radial distributions, configuration interaction, quantum dots

## Abstract

Using an ab initio, time-dependent calculational method, we study the non-linear dynamics of a two-electron quantum dot in the presence of ultrashort Thz laser pulses. The analysis of the contribution of the various partial waves to two-electron joint radial and energy distribution patterns revealed strongly correlated electron ejection channels. In the double-ionization process, regardless of the photon energy, the two-electron wave packets are born and remain concentrated until the pulse’s peak; at later times, and depending on the photon energy of the field, distinctly different patterns emerge. Our calculations also showed the gradual transition of the radial and energy patterns from a single-peak to a doubly peaked structure, associated with the direct and the sequential double-ionization mechanisms, respectively.

## 1. Introduction

The study of the optical properties of semiconductor quantum dots (QDs) is of paramount importance in the research domains of fundamental theory and applications for quantum information processing, solar energy harvesters, optoelectronics, etc. [1]. Due to the high degree of flexibility of QD design, it is possible to artificially control the transport and optical properties, in contrast to quantum structures of a similar size such as atoms, molecules, and bulk materials [2,3,4,5,6]. The non-linear properties of QDs (e.g., GaAs, CdSe, etc.) are a relatively unexplored scientific area with their study complicated by the highly correlated nature of their electronic structure. On the experimental side, it has been demonstrated that the use of laser pulses in the infrared and terahertz regimes allows direct measurement of their electronic and geometrical properties [7,8,9,10,11,12,13,14]. In particular, the dependence of the optical properties of colloidal PbS QDs was demonstrated in [7], while in a pioneering experiment in [13], photoelectron spectroscopy measurements of CdSe QDs in the gas phase were reported.

The vast majority of the corresponding theoretical studies of QDs’ optical properties have assumed a weak external electromagnetic field [15] and very often rely on the electronic structure of a considerably simplified system, often ignoring interelectronic correlations, as, for example, in Fominykh et al., who studied the QD’s one-photon double-emission process in a harmonic potential well [16,17]. These studies are rendered invalid for intense and ultrashort Thz laser fields since excitation/ionization with the QDs may proceed non-linearly with the applied field; for the same reason, the electronic structure of the QD, both the bound and continuum part, plays a decisive role in the system’s dynamics, which does not allow oversimplification of the QD/laser system theoretical description.

In an effort to study the fundamental optical properties of simple quantum dots, we developed an ab initio theoretical formulation for the description of excitation and ionization mechanisms in two-electron quantum dots. The theoretical description of the QD electronic structure is based on a single-band effective mass model with a spherical Gaussian potential well, adapted for semiconductor nanocrystals [18,19,20]. Within this effective mass model, we extended the theory to include interelectronic interactions to represent the electronic structure more accurately from existing theoretical results [21,22,23,24,25]; this allowed us to calculate the ionization cross-sections and yields, as well as radial, angular, and kinetic energy electron distributions [26,27,28]. As is shown in these works, interelectronic interactions may radically affect the observed radial, kinetic energy, and angular patterns of the ejected electrons (collectively, *ionization* patterns). In particular, the study of the ionization process may provide the means to clarify the often competitive role of an external time-dependent electric field and the static Coulombic field due to interelectronic interactions. Generally, while the external laser field is certainly the primary agent to trigger ionization, it is not the exclusive factor that affects the final states of the QD; the interelectronic interactions may also play an important and decisive role under certain circumstances. The absorption of two photons may lead directly to a doubly ionized system, QD${}^{+2}$, and two outgoing electrons; this is known as direct two-photon double-ionization (direct TPDI) (see Figure 1). However, an essential change may occur when the photon energy of the pulse becomes higher than the ionization threshold of the singly ionized QD${}^{+}$. In this case, TPDI may proceed initially by the absorption of one photon from the neutral QD, leading to QD${}^{+}$, followed by the absorption of one further photon by QD${}^{+}$, leading to QD${}^{2+}$ and two free electrons; this is the sequential TPDI mechanism. Therefore, for both direct or sequential TPDI, the final products are the same, but the underlying ionization mechanism proceeds with maximized electron–electron interaction for the direct TPDI, but minimized for sequential TPDI [27,28].

While the study of these ionization patterns does provide useful information and is accessible by direct experimental observation, a deeper insight can also be obtained by the study of their dynamics. Therefore, we may examine how these final states are actually reached as a result of the laser field and the interelectronic interactions, in other words: How do they evolve in time? From our past studies of the two-photon double-ionization of QDs, we clearly identified the conditions leading to the two different ionization mechanisms.

In the present work, we studied the two-photon double-ionization mechanisms of two-electron QDs from the viewpoint of a time-dependent two-electron spatial wave packet. We clarified the TPDI mechanisms and the individual roles of the laser field and the interelectronic interactions based on the calculated radial and energy distribution patterns, analyzed in terms of partial waves in combination with the time elapsed between the two photon absorptions, in a time-resolved manner. We left out the time evolution of angular correlations as it is not required for the essential conclusions of the present study; nevertheless, the interested reader may read [29,30] or [31,32,33], in the context of two-electron atomic systems.

We should also mention here that the discussion applies regardless of the QD’s semiconductor material provided the proper scale of the pulse’s photon energy is also considered to match the QD’s electronic structure. For our study, we took a QD model by choosing the values of the electronic effective mass, $m_{e}^{*}$, and dielectric constant, ε, corresponding to a CdSe semiconductor type, and then, we chose its size by choosing the depth and the radius of the Gaussian potential [13,24]; accordingly, the properties of a GaAs QD type may be studied by choosing $m_{e}^{*}=0.067$ and ε=12.4, as in [25].

The current Section 1 serves to introduce the context of our work and to provide a preliminary physical insight into the problem in hand. In Section 2, we present the theoretical formulation for the calculation of the two-electron radial and energy spectra; since the formulation has been presented in detail elsewhere [26,27,28], we present only the formulas suited to the present study. In Section 3, we discuss the time evolution of the ionization patterns in the TPDI process, as well as their variation as the photon energy crosses the direct and sequential double-ionization photon energy threshold (SDI/DDI threshold). Finally, in Section 4, we conclude by summarizing our main findings and perspectives.

## 2. Theoretical Formulation

In the following, the formulation for the ab initio calculation of the two-electron joint radial and energy probability distribution is developed. Briefly, we calculated the two-electron wave function at the end of the laser pulse and then extracted the desired information for the evaluation of observables via the various coefficients of the system. The method is based on a non-perturbative solution of the time-dependent Schrödinger equation (TDSE) for the QD interacting with a linearly polarized pulse. The computational code was developed in house and has been thoroughly tested and used over the years, albeit in the context of atomic physics by studying the response of atomic systems in the presence of laser fields [26,27,28]. As in these works, the method was presented in detail (and the references therein), here, we discuss only briefly the main theoretical formulation and emphasize more the formulas that are necessary for our purposes.

### 2.1. Quantum Dot Model Potential

We start with the model that describes a spherical two-electron quantum dot built from a narrow band gap semiconductor crystal of approximate radius $r_{q}$. Our starting point is to model the two-electron QD as
(1)$${\hat{H}}^{Q}=-\frac{\hbar^{2}}{2m_{e}^{*}}\nabla_{i}^{2}+V_{QD}\left(r_{i}\right)+\frac{e^{2}}{4\pi \varepsilon }\frac{1}{|r_{1}-r_{2}|},$$
where $m_{e}^{*}$ and ε represent the electronic effective mass and the dielectric constant, which are uniform in space. The next step is to re-scale the physical parameters and convert the Hamiltonian to a convenient form. We introduced the relative mass and dielectric parameter, μ and κ, and scaled as $m^{*}=\mu m_{e}$ and $\varepsilon =\kappa \varepsilon_{0}$, where $m_{e}$ and $\varepsilon_{0}$ are the electronic mass and the vacuum’s dielectric constant. Following this approach, the equations that follow are presented in a scaled atomic unit system (see Table 1), to suit the order of magnitude of the system’s constants, $m_{e}^{*},\varepsilon$ [26].

Next, we chose a potential model for the QD; it is worth noting here that the electronic structure of a QD compared to an atomic system differs in the central potential, V(r). Essentially, we changed $V\left(r\right)=-Z_{nuc}/r$ to a suitable potential for the QD under study ($Z_{nuc}$ is the atomic number). Here, the QD was modeled by a Gaussian potential with a width parameter of $r_{q}=3.2$ nm (1.21 s.a.u) and a depth of $V_{0}=-542.2$ meV (−5 s.a.u):(2)$$V_{QD}\left(r\right)=-V_{0}e^{-\ln2{\left(\frac{r}{r_{q}}\right)}^{2}}.$$

The full details of the model we chose to calculate the QD structure are discussed in [26].

These parameters give for the ground state energy of the neutral quantum dot, QD, $E_{0}=-365.2$ meV, and for the single-electron quantum dot, QD${}^{+}$, $E_{1}=-232.9$ meV. These values are relative to the double-ionization threshold, which was set as the zero of the energy axis $\left(E_{2}=0\right)$. In Table 2, we give a few of the calculated energies of the neutral and singly ionized QD.

The value of the QD${}^{+}$ ionization potential ($I_{1}=E_{2}-E_{1}=232.9$ meV) determines the photon energy regimes for the two distinct TPDI mechanisms to occur (SDI/DDI threshold). Therefore, in the following, the QD has a fixed size, but we varied the photon energy of the laser pulse in order to investigate both TPDI mechanisms.

Having chosen the QD model potential, we proceed with the formulation of our approach.

### 2.2. Theoretical Formulation of the Quantum Dot Structure

We initially constructed the two-electron singlet antisymmetric uncorrelated basis, with sharp angular momentum values, *L*, as the solutions of the eigenvalue problem of the zero-order Hamiltonian, ${\hat{H}}_{0}^{Q}={\hat{h}}_{1}^{Q}+{\hat{h}}_{2}^{Q}$, where the one-electron QD${}^{+}$ Hamiltonian is given by:(3)$${\hat{h}}_{i}^{Q}\left(r_{i}\right)=-\frac{1}{2}\nabla_{i}^{2}+V_{QD}\left(r_{i}\right),$$
where *i* indexes the two electrons and $V_{QD}\left(r_{i}\right)$ is the chosen QD potential model (Equation 2). Following a standard angular momentum coupling algebra, these solutions in a spherical coordinate system, with its origin placed at the quantum dot core, may be expressed as: $$\phi_{n_{1}l_{1}n_{2}l_{2}}^{L}\left(r_{1},r_{2}\right)=A_{12}\frac{P_{n_{1}l_{1}}\left(r_{1}\right)}{r_{1}}\frac{P_{n_{2}l_{2}}\left(r_{2}\right)}{r_{2}}Y_{l_{1}l_{2}}^{L0}\left(\Omega_{1},\Omega_{2}\right),$$
where $Y_{l_{1}l_{2}}^{LM_{L}=0}\left(\Omega_{1},\Omega_{2}\right)$ are the bipolar spherical harmonics, containing the angular momentum coupling coefficients (Clebsch–Gordon coefficients) and $A_{12}$ is the antisymmetrization operator, which acts to exchange the coordinates of the two electrons. The radial orbitals $P_{n_{i}l_{i}}\left(r_{i}\right),\;i=1,2$ were found from the solutions of the radial eigenvalue problem of the QD${}^{+}$ Hamiltonian, namely
(4)$$\left[-\frac{1}{2}\frac{d^{2}}{dr^{2}}+\frac{l(l+1)}{2r^{2}}-V_{0}e^{-\ln2{\left(\frac{r}{r_{q}}\right)}^{2}}\right]P_{nl}=\epsilon_{nl}P_{nl}\left(r\right).$$

The purely radial functions, $P_{nl}\left(r\right)$, were obtained numerically by expanding on a piecewise polynomial basis (B-splines); the choice of this particular basis is dictated by its numerical ability for representing continuum states, a property that is of importance in the particular case where ionized states are involved [29,30,31]. The QD${}^{+}$ system was assumed to be confined in a sphere of radius *R*, much larger than the quantum dot size (R≫1s.a.u). Within our particular approach, we implemented the so-called *fixed-boundary conditions*, which require the wavefunctions to strictly vanish at the origin and the boundaries, $P_{nl}\left(R\right)=0$. As a result of this requirement, the QD${}^{+}$ eigenstates of Equation (Equation 3) are discretized, allowing the bound and continuum spectrum to be represented by negative and positive energy orbitals, respectively, subject to unity normalization. In this case, the index *n* of $P_{nl}\left(r\right)$ takes integer values, n=1,2,.. and the sign of $\epsilon_{nl}$ determines whether we have an exponentially decaying (negative) or an oscillatory (positive) radial orbital.

Having completed the numerical calculation of the partial-wave radial orbitals of QD${}^{+}$, the neutral QD Hamiltonian is modeled by
(5)$${\hat{H}}^{Q}={\hat{H}}_{0}^{Q}+\frac{1}{|r_{1}-r_{2}|},$$
with the second term on the right-hand-side representing the inter-electronic interaction potential. The eigenvalue problem to be solved is the time-independent Schrödinger equation:(6)$${\hat{H}}^{Q}\Phi_{EL}\left(r_{1},r_{2}\right)=E_{L}\Phi_{EL}\left(r_{1},r_{2}\right),$$
where $\Phi_{EL}\left(r_{1},r_{2}\right)$ are the two-electron eigenstates of ${\hat{H}}^{Q}$. Following the interaction of helium with a linearly polarized laser, only the $M_{L}=0$, singlet symmetry states ($S,M_{S}$) = (0,0) are excited, since the total magnetic quantum number and the initial spin state do not change, so we considered these states only.

For the solution of Equation (Equation 6), a configuration interaction (CI) method was employed where the QD eigenstates, $\Phi_{EL}$, were expanded on the zero-order two-electron basis $\phi_{n_{1}l_{1}n_{2}l_{2}}^{L}\left(r_{1},r_{2}\right)$ as
(7)$$\Phi_{NL}\left(r_{1},r_{2}\right)=\sum_{n_{1}l_{1}n_{2}l_{2}}v_{n_{1}l_{1}n_{2}l_{2}}^{NL}\phi_{n_{1}l_{1}n_{2}l_{2}}^{L}\left(r_{1},r_{2}\right).$$

Since the expansion is on discretized orbitals, the resulting two-electron CI states are also discretized along with their associated energy, *E*; for this reason, it is more consistent to use a discretized notation for the states and the energy, thus using $E_{L}\to E_{NL}$ and $\Phi_{EL}\to \Phi_{NL}$, with N=1,2,⋯.

Substituting Equation (Equation 7) into Equation (Equation 6), followed by projection over $\phi_{n_{1}l_{1}n_{2}l_{2}}^{L}\left(r_{1},r_{2}\right)$, converts it to a matrix equation, which, upon diagonalization, retrieves the eigenenergies $E_{NL}$ and the CI amplitude coefficients $v_{n_{1}l_{1}n_{2}l_{2}}^{NL}$. For a two-electron system, there are three characteristic energies, namely the energies of the ground state, the single-ionization threshold, and the double-ionization threshold, denoted here by $E_{0}$, $E_{1}$, and $E_{2}$, respectively. Conventionally, the latter energy was set to $E_{2}\equiv 0$ (double-ionization threshold energy). Within the approximations introduced in the numerical computation of these discretized energies, it turns out that a two-electron eigenstate with $E_{NL}<E_{1}$ is of a bound character and, thus, represents bound quantum dot states, while states with energies $E_{1}<E_{NL}<0$ represent singly ionized quantum dots; finally, the numerical states with $E_{NL}>0$ may represent both singly and doubly ionized quantum dots with one electron ejected or two electrons ejected, respectively [32].

### 2.3. TDSE of Quantum Dot in the Laser Field

Once the electronic structure of the quantum dot has been solved for, its electronic dynamics under the influence of an intense and ultra-short linearly polarized laser pulse can be solved for. This amounts to solving the time-dependent Schrödinger equation (TDSE) for the combined quantum dot laser system. In this case, the semiclassical TDSE that describes the quantum dot in the presence of a laser field is given by:(8)$$ı\frac{\partial }{\partial t}\Psi \left(r_{1},r_{2},t\right)=\left[{\hat{H}}^{Q}+\hat{D}\left(t\right)\right]\Psi \left(r_{1},r_{2},t\right),$$
where $\Psi (r_{1},r_{2},t)$ is the time-dependent wavefunction and D(t) describes the external laser interaction dipole potential in the Coulomb gauge:(9)$$D\left(t\right)=\frac{1}{c}A\left(t\right)\hat{z}\cdot \left({\hat{p}}_{1}+{\hat{p}}_{2}\right),$$
where $\hat{z}$ is the unit vector along the z-axis and $p_{1}$ and $p_{2}$ are the electron momenta. A(t) is the amplitude of the electromagnetic potential field related to the electric field of the pulse by $E\left(t\right)=-\dot{A}\left(t\right)/c$. Note that the long-wavelength approximation was taken into account in the expressions above. In this work, we chose the amplitude envelope to be
(10)$$A\left(t\right)=A_{0}\sin^{2}\left(\frac{\pi t}{\tau_{p}}\right)\sin\;\omega t,\;0\leq t\leq \tau_{p}.$$
where ω is the carrier frequency. The use of a squared sinusoidal envelope satisfies the requirements that the envelope varies slowly with respect to the carrier period and rises and falls to zero; $\tau_{p}$ is the laser pulse duration, related to the field period ($T_{0}=2\pi /\omega$) by $\tau_{p}=n_{c}T_{0}$, where $n_{c}$ is the number of field cycles in the pulse.

A spectral expansion of the solution of Equation (Equation 8) in terms of the two-electron CI eigenstates with a time-dependent amplitude as
(11)$$\Psi \left(r_{1},r_{2},t\right)=\sum_{NL}C_{NL}\left(t\right)\Phi_{NL}\left(r_{1},r_{2}\right).$$
allows for the interpretation of $|C_{NL}{\left(t\right)|}^{2}$ to be the population of the state $\Phi_{NL}$, since it represents the probability of observing the system in state $\Phi_{NL}\left(r_{1},r_{2}\right)$ at time *t*.

Formally, the substitution of this latter expansion into Equation (Equation 8) and multiplying from the left by $\Phi_{N^{\prime }L^{\prime }}^{*}\left(r_{1},r_{2}\right)$, followed by spatial integration over the entire coordinate space, transforms the TDSE into a set of coupled ordinary differential equations:(12)$$ı{\dot{C}}_{NL}\left(t\right)=E_{NL}C_{NL}\left(t\right)+\sum_{{}_{N^{\prime }L^{\prime }}}D_{NL;N^{\prime }L^{\prime }}C_{N^{\prime }L^{\prime }},$$
where $D_{NL;N^{\prime }L^{\prime }}$ are the dipole matrix elements between the CI states $\Phi_{NL}$ and $\Phi_{N^{\prime }L^{\prime }}$. Standard angular momentum algebra allows for the expression of the two-electron dipole matrix elements in terms of the $v_{n_{1}l_{1}n_{2}l_{2}}^{NL}$ CI coefficients and the one-electron dipole matrix elements; the latter are calculated numerically given the $P_{nl}\left(r\right)$ radial functions [26]. Thus, solving the TDSE amounts to calculating only the two-electron dipole matrix elements and integrating Equation (Equation 12) to find the time-dependent coefficients, $C_{NL}\left(t\right)$.

### 2.4. Radial and Kinetic Energy Distributions

At this point, we are in the position to proceed with the main subject, which is finding the electron’s radial and kinetic energy distributions, and in accordance with our framework, these are expressed in terms of the calculated time-dependent coefficients, $C_{NL}\left(t\right)$, along with the CI coefficients, $v_{n_{1}l_{1}n_{2}l_{2}}^{NL}$. Since these derivations are more specialized and naturally of a technical character we delegated the relevant algebra in the associated Supplemental Text for further consideration.

We can express the two-electron joint photoelectron spectrum (JPES) in terms of configuration interaction coefficients and time-dependent coefficients by
(13)$$P_{l_{1}l_{2}}\left(\epsilon_{1},\epsilon_{2};t\right)=\sum_{L}{\left|\sum_{N}v_{n_{1}l_{1};n_{2}l_{2}}^{NL}C_{NL}\left(t\right)\right|}^{2},$$
where $v_{n_{1}l_{1};n_{2}l_{2}}^{NL}$ are the CI coefficients in Equation (Equation 7) and $\epsilon_{1},\epsilon_{2}$ the kinetic energies associated with the $P_{n_{1}l_{1}}\left(r\right)$ and $P_{n_{2}l_{2}}\left(r\right)$ radial orbitals, respectively.

The joint photoelectron radial distribution (JPRD) can be found by evaluating the square of the time-dependent wavefunction, followed by integration over the angular coordinates:(14)$$P_{r}\left(r_{1},r_{2},t\right)=\sum_{Ll_{1}l_{2}}{\left|\chi_{l_{1}l_{2}}^{\left(L\right)}\left(r_{1},r_{2},t\right)\right|}^{2},$$
where
(15)$$\chi_{l_{1}l_{2}}^{\left(L\right)}\left(r_{1},r_{2},t\right)=\sum_{Nn_{1}n_{2}}C_{NL}\left(t\right)v_{n_{1}l_{1};n_{2}l_{2}}^{\left(NL\right)}\rho_{n_{1}l_{1};n_{2}l_{2}}\left(r_{1},r_{2}\right),$$
and
$$\rho_{n_{1}l_{1};n_{2}l_{2}}\equiv \frac{1}{\sqrt{2}}\left[P_{n_{1}l_{1}}\left(r_{1}\right)P_{n_{2}l_{2}}\left(r_{2}\right)+P_{n_{1}l_{1}}\left(r_{2}\right)P_{n_{2}l_{2}}\left(r_{1}\right)\right].$$
which represents the probability distribution of finding the two electrons with angular momenta $\left(l_{1},l_{2}\right)$ at radial distances $r_{1}$ and $r_{2}$ from the core of the quantum dot. Note that, under the given conditions, the radial distribution is symmetrical to the exchange $r_{1}↔r_{2}$, meaning that $P_{r}\left(r_{1},r_{2},t\right)=P_{r}\left(r_{2},r_{1},t\right)$.

## 3. Results and Discussion

The quantum dot electronic structure was calculated using the two-electron wavefunction expansion in Equation (Equation 7) with electronic configurations $\left(n_{1}l_{1};n_{2}l_{2}\right)$ and L=0−3; if the ground state is of singlet spin symmetry, the interaction with a linearly polarized pulse can only excite singlet spin symmetry states; so the configuration channels included in the calculations satisfy $l_{1}+l_{2}+L=$
*even*. The box radius chosen for the calculation was R=60 s.a.u. (∼160 nm). We kept all configurations with both $l_{1},l_{2}$ up to three, which are adequate for the chosen laser parameters. We checked that our results did not change in any significant way by incorporating additional partial waves $\left(l_{1},l_{2}\right)$ or by further increasing the box radius.

For the calculations, we used linearly polarized laser pulses with central carrier frequencies between ω = 195 and 308 meV, at a peak intensity $I_{0}=6.4\times 10^{5}$ W/cm${}^{2}$. As the number of cycles was kept constant ($n_{c}=12$), the pulse durations ranged between 0.16 and 0.25 ps.

### 3.1. Time Evolution of the Radial Wave Packets

As a result of the importance of the electron–electron correlations between the sequential and direct DI regimes, we wanted to examine what quantitative differences showed up in the radial probability patterns during the double-ionization process. For this, we examined the time evolution of the corresponding radial distributions as given by Equation (Equation 14) with the summations restricted to $N,n_{1},n_{2}$ such that $E>0,\epsilon_{1},\epsilon_{2}>0$.

For ω<232.9 meV, ionization proceeds via the direct channel, namely the absorption of two photons by the neutral quantum dot, leading to excitation states where the two electrons leave the core region simultaneously. A representative set of snapshots of the radial distribution of the dominant (p,p) wave packets for an ω=195 meV pulse is shown in the left column of Figure 2. The snapshots correspond to times where the field vanishes and the interpretation of the time-dependent coefficients as field-free probability amplitudes for the quantum dot eigenstates is well justified, as for these times, the system’s Hamiltonian coincides with the field-free quantum dot Hamiltonian.

Initially, the generated two-electron wave packets remain in the proximity of the core at distances no more than 11 nm (4.16 s.a.u.), until up to the pulse’s peak; the shape of the distribution remains concentrated about the $r_{1}=r_{2}$ diagonal. It is expected that the e–e correlation interactions have a maximal contribution to the experimental observables due to the proximity of the two electrons (since the correlation interaction potential vanishes with the inverse of the interelectronic distance).

Turning now to the sequential regime, for ω>232.9 meV, the main contribution to the double-ionization process comes from electrons generated at times that are so distinctly different that the second electron is essential ejected from QD${}^{+}$ in its ground state; in the present case, the time taken for the residual quantum dot (following one-photon absorption) to relax to the QD${}^{+}$ ground state is about 0.01 ps), which means about 0.01/0.013∼4.5 cycles for a 308 meV pulse (the pulse period is 0.013 ps). In the corresponding radial distributions, shown in the right column of Figure 2, the formation of the asymmetric radial distribution is due to the enhancement of double-ejection channels originating from a further photon absorption some time later than the instant of the first photon absorption.

From these figures, we can infer the different excitation mechanisms of the double-ejection events in the sequential and the direct DI regimes. In the sequential regime (308 meV pulse), first, a small portion of the wave packet becomes excited, consisting of channels with both electrons in the continuum, traveling along the $r_{1}=r_{2}$ diagonal; the largest ionization portion corresponds to channels with only one electron ejected (not shown in the plots). However, soon after (past the pulse’s peak), the presence of this larger portion overwhelms the double-ionization distribution, since it contributes an additional channel by further absorption of the second photon by the QD${}^{+}$. On the other hand, in the direct DI regime (195 meV pulse), the sequential ionization mechanism is absent at later times; thus, the electronic wave packet remains concentrated along the $r_{1}=r_{2}$ diagonal, throughout the field’s lifetime. In both ionization mechanisms, sequential or direct, the wave packets appear to depart from the core region well past the peak of the pulse and experience the natural broadening of free-moving wave packets; however, in the direct mechanism, the e–e interactions may cause further distortion of the wave packets via population redistribution among the various partial waves $\chi_{l_{1},l_{2}}\left(r_{1},r_{2}\right)$.

### 3.2. Time Evolution of the Energy Distributions

Similar conclusions may be reached by the observation of the corresponding joint energy distribution patterns, shown in Figure 3 and Figure 4 for the 308 meV and 195 meV pulses. In these figures, we plot the results of the calculations with only the DI channels included in Equation (Equation 13), with $E,\epsilon_{1},\epsilon_{2}>0$. Initially, the excitation is rather evenly distributed among the partial waves of the DI wave packets. For example, the inspection of the time-ordered plots of Figure 3 for the 308 meV pulse is consistent with the observation that the sequential DI is overwhelmed by channels where one of the electrons is ejected from the quantum dot, with energy $\epsilon_{1}=E_{0}+\omega -E_{1}∼176$ meV (1.62 s.a.u.), and the other from QD${}^{+}$, with energy $\epsilon_{2}=E_{1}+\omega -E_{2}∼75$ meV (0.69 s.a.u.); however, it is only at the later stages of the interaction with the field that this asymmetrical energy distribution shows up as the dominant feature.

For the 195 meV pulse corresponding to the direct DI regime, a persistent, evenly distributed pattern is observed throughout the pulse’s duration. We should note here that, for the (p,p) channel in the direct DI regime, the two electrons may become ejected with comparable energies, but with their momenta pointing in the same or opposite directions (back-to-back ejection); in the latter case, the two electrons evolve practically independently with each other, and their interelectronic interactions are minimized.

### 3.3. Transition from Direct to Sequential Regime

In the final plots (cycle 12) of Figure 2, Figure 3 and Figure 4, we can identify the characteristic features of the direct and sequential DI radial patterns, represented by the 195 meV and 308 meV pulses: a single-peak versus two-peak radial pattern for the 195 meV and 308 meV, respectively. In the latter case, as mentioned, the two peaks correspond to the ejection of the electrons from the neutral (∼176 meV) and the singly ionized QD${}^{+}$ (∼75 meV). These two photon energies are well distant from the DI threshold (232.9 meV) and, as such, may be considered as “representative” cases of the sequential and the direct DI mechanisms.

Taking into account that both mechanisms do not occur independently of each other, alongside the existence of the bandwidth of the pulse, the question is raised about what form these patterns will take for pulses with photon energies in between these two characteristic energies and closer to the DI threshold. To answer this question, we calculated the radial distribution for photon energies 205 meV, 214 meV, 223 meV, 233 meV, 242 meV, 252 meV, 270 meV, and 289 meV and plotted the results in Figure 5 along with those of the 195 meV and 308 meV pulses (top left and bottom right plots of the same figure).

From these plots, it is clear that the final distributions do not distinctly belong to each of these two regimes; we observed the gradual deformation of a single-peak pattern along the diagonal line to a double-peak asymmetric pattern as the photon energy crosses the 232.9 meV DDI/SDI threshold, finally leading to the two-peak structure for the higher photon 308 meV. It is also worth noting that the double-peak structures are aligned with the $r_{1}$ and $r_{2}$ axes, which suggests that the double-ejection occurs with both of the electrons moving outwards, but with different speeds. For pulses with photon energies closer to the 232.9 meV threshold, the second electron is excited from broadband pulses, and contributions from both the sequential and direct DI mechanism are present; for example, the 12-cycle, 233 meV pulse has a bandwidth of 30 meV.

A more compact view of the effects of the two ionization mechanisms can be obtained via the single-electron radial distribution, obtained by the integration of the two-electron radial distribution along one of the radial coordinates:(16)$$P_{r}\left(r_{1}\right)=\int_{0}^{\infty }dr_{2}\;P_{r}\left(r_{1},r_{2},\tau_{p}\right).$$

Due to the symmetry of the two-electron radial distribution, integration over the other radial coordinate would result in identical values, so that $P_{r}\left(r_{1}\right)=P_{r}\left(r_{2}\right)$. The results of the calculations are shown in Figure 6. From this plot, it is again evident that the single peak in the radial distance corresponds to electrons originating from the direct DI mechanism with similar energies, while the doubly peaked structure represents bursts of electrons ejected sequentially and with different kinetic energies in accordance with Equation (Equation 17).

In Figure 7, we provide the joint kinetic energy distributions with the same range of photon energies. In agreement with the conclusions drawn from the two-electron radial distributions, we again observed the gradual deformation from a symmetric single-peak pattern to a doubly peaked asymmetric one. Clearly, the peaks in the sequential spectra eventually tend to satisfy:(17)$$\epsilon_{1}≃E_{0}+\omega -E_{1},\;\epsilon_{2}≃E_{1}+\omega -E_{2},$$
where one of the electrons is likely to obtain more energy than the other as a consequence of the sequential nature of the ionization process. Based on our observations thus far, the latter may be envisaged when the two-photon absorptions occur at times that differ by an amount that is larger than the relaxation time of QD${}^{+}$ (of the order of $\tau_{r}∼2\pi /I_{1}∼0.01$ ps). At this time, the primary two-electron wave packet, generated by the first-photon absorption, has evolved to a state resembling a QD${}^{+}$ bound orbital and an outgoing wave packet $\psi_{t}^{+}∼\phi_{nl}\left(t\right)\phi_{\epsilon_{1}}\left(t\right)$, peaked at $\epsilon_{1}=E_{0}+\omega_{1}-E_{1}$. The subsequent time evolution of this two-electron wave packet depends on whether the absorption of the second photon (with energy $\omega_{2}$) suffices to promote the bound electron to a continuum state. For example, if $\omega_{2}<232.9$ meV, further absorption cannot lead to a transition QD${}^{+}\left(1s\right)\to$ QD${}^{+}\left(\epsilon p\right)$, but it may proceed via above-threshold ionization (ATI) by further exciting the continuum wave packet $\phi_{\epsilon_{1}}\left(t\right)$, leading to $\psi_{t}^{+}∼\phi_{1s}\left(t\right)\phi_{\epsilon_{1}+\omega_{2}}\left(t\right)$, eventually corresponding to a singly ionized quantum dot, QD${}^{+}$. On the other hand, if $\omega_{2}>232.9$ meV, it is much more probable for the transition $\phi_{nl}\left(1s\right)\to \left(\epsilon_{2}p\right)$ to occur, leading to a wave packet of the type $\psi_{t}^{2+}∼\phi_{\epsilon_{2}p}\left(t\right)\phi_{\epsilon_{1}p}\left(t\right)$ with $\epsilon_{2}∼E_{1}+\omega_{2}-E_{2}$. In the above, $\omega_{1}$ and $\omega_{2}$ may differ, but are restricted to lie within the bandwidth of any given pulse. From this discussion, for pulses with photon energies ω>232.9 meV and a duration longer than the relaxation time of the residual QD${}^{+}$ to its ground state (∼0.01 ps), the sequential DI mechanism leads to a kinetic energy spectrum peaked at $\epsilon_{1}$ and $\epsilon_{2}$, which, for moderate intensities, have a width mainly determined by the pulse’s bandwidth. Both electrons have l=1 angular momentum and, therefore, are (p,p) channels, characterized by the radial probability distribution $|\chi_{11;L}{|}^{2},\;L=0,2$. It is important to point out that for ω>232.9 meV, direct DI, corresponding to the “simultaneous” absorption of photons, is of course still energetically possible and, thus, as observed, is a contribution to the two-electron wave packet originating from this mechanism, albeit not representing the dominant contribution.

## 4. Conclusions

By directly solving the time-dependent, full-dimensional, two-electron Schrödinger equation for a spherical two-electron quantum dot in the field of a laser pulse, we investigated the time evolution of the generated radial wave packets during the two-photon double-ionization process. We carried out a systematic analysis of the joint radial and energy distributions of the two ejected electrons to elucidate some aspects of the role of electron correlations in the two-photon double-ionization process. The investigation included pulses with photon energies that favor either a direct or a sequential TPDI mechanism.

More specifically, we provided the time evolution of the radial and kinetic energy ejection patterns during the interaction of the QD with the external laser field. Moreover, we investigated the gradual deformation of these distributions from the single-peak structure associated with the direct TPDI mechanism to the two-peak structure deep in the regime of sequential TPDI.

The present space–time description provides an enhanced view of the two-photon double-ionization processes, complimentary to the corresponding studies focusing exclusively on the final-state distributions. In view of the potential applications of quantum dot systems in optical and photonic applications [34,35,36], it is our intention to work further in this direction by studying more complex quantum dot systems of experimental interest.

## Figures and Tables

**Figure 1 materials-16-01405-f001:**
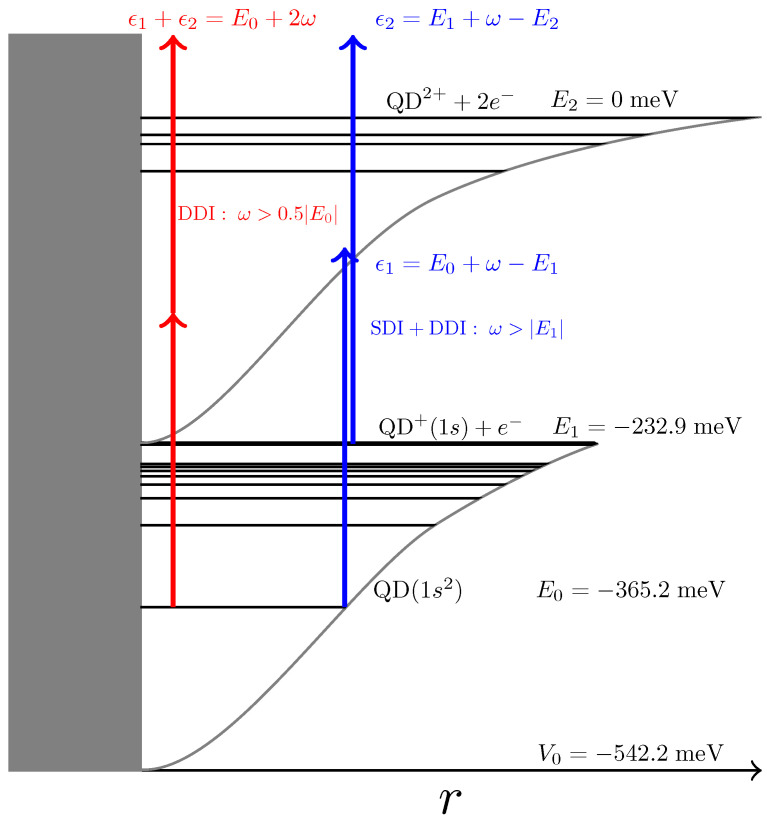
Sketch depicting the two TPDI mechanisms, sequential (blue) and direct (red). The energies correspond to a Gaussian-modeled QD with parameters given in the text.

**Figure 2 materials-16-01405-f002:**
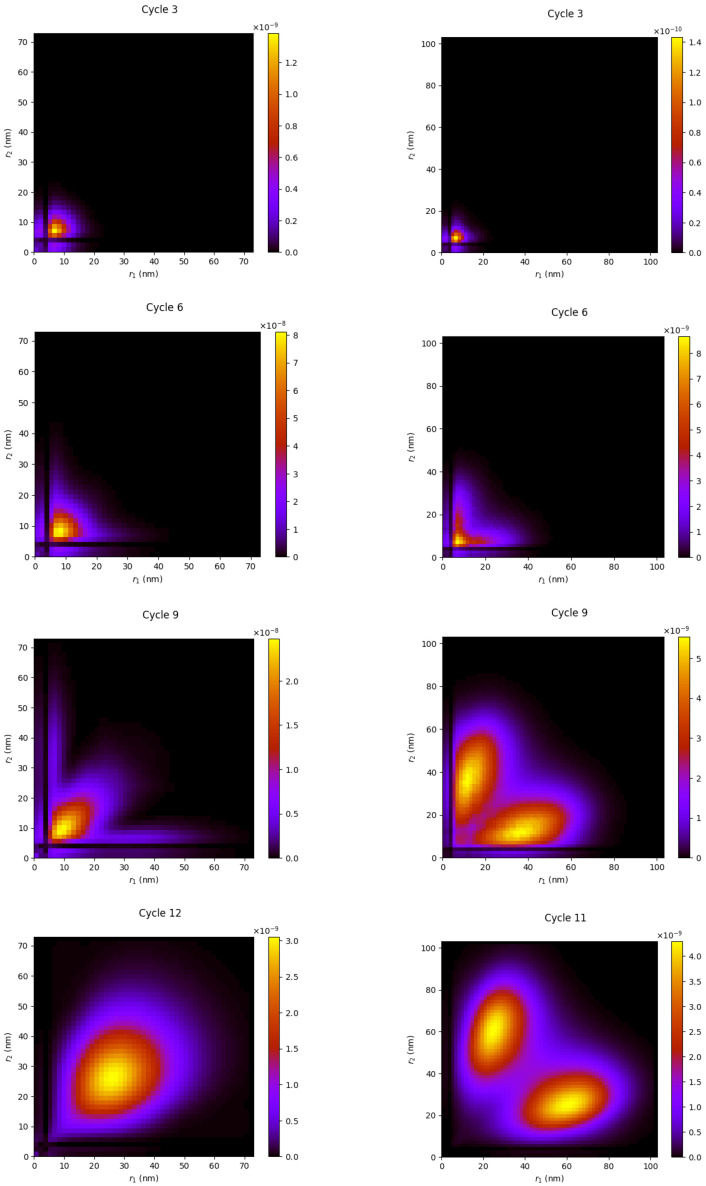
The side by side comparisons of the time evolution of radial distributions for a 195 meV pulse (**left** column, NSDI) and a 308 meV pulse (**right** column, SDI), for the dominant (p, p) partial wave.

**Figure 3 materials-16-01405-f003:**
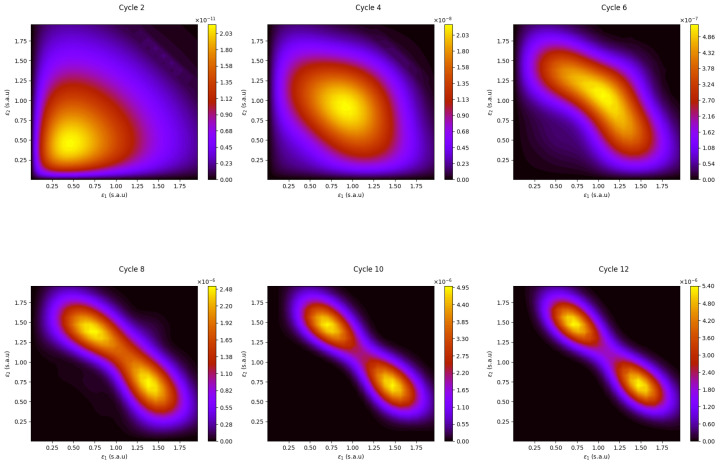
The time evolution of the joint PES for the 308 meV pulse with parameters as in Figure 2, right plot.

**Figure 4 materials-16-01405-f004:**
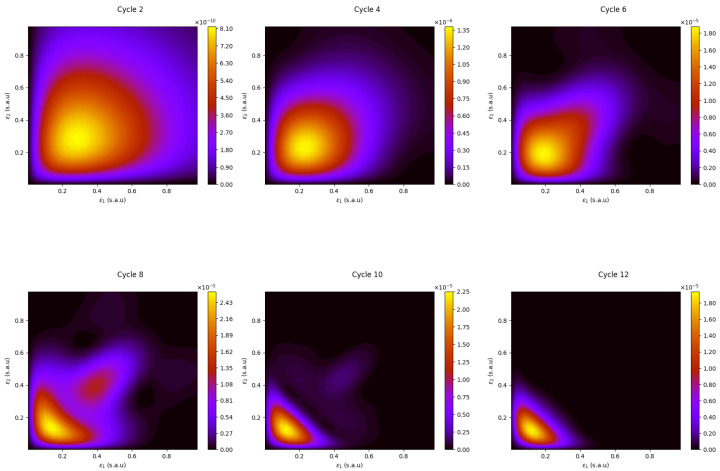
The time evolution of the joint PES for the 195 meV pulse with parameters as in Figure 2, left plot.

**Figure 5 materials-16-01405-f005:**
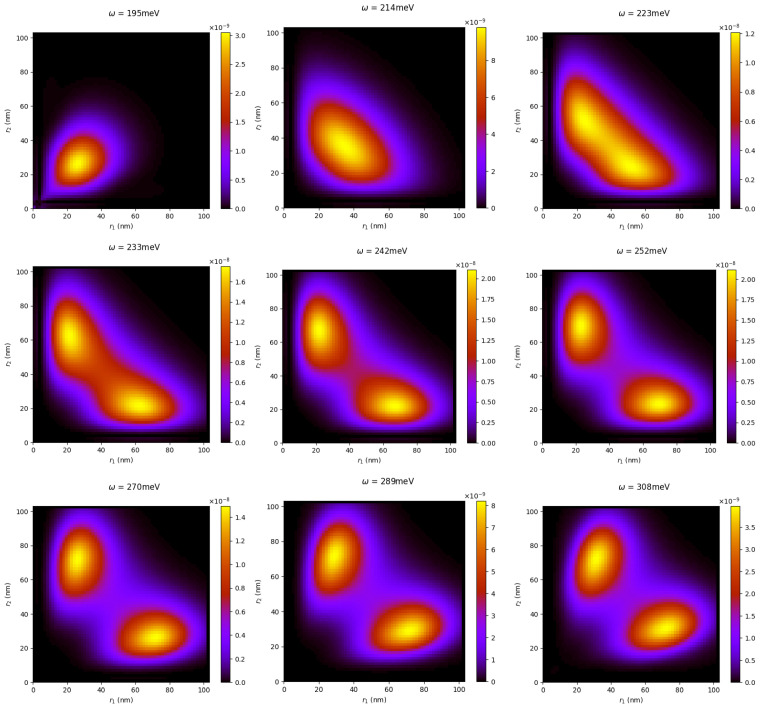
Clearly visible in the radial distribution patterns is the transition from the direct to the sequential double-ionization regime as the photon energy varies from 195 meV to 308 meV for a 12-cycle pulse.

**Figure 6 materials-16-01405-f006:**
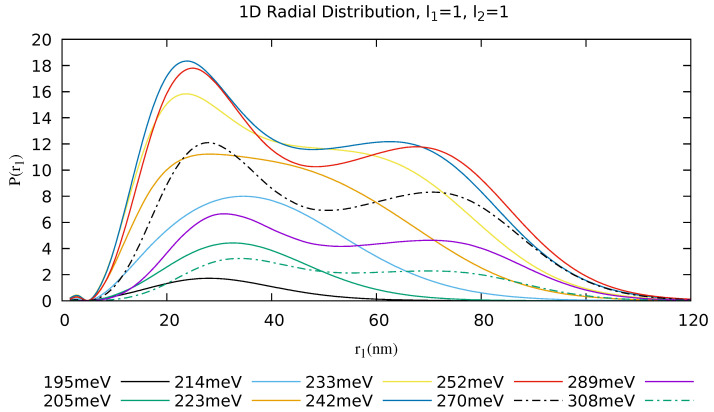
Two-dimensional radial distributions integrated over one radial coordinate demonstrates clearly the presence of peaks in the radial distributions (ω = 195 meV–308 meV).

**Figure 7 materials-16-01405-f007:**
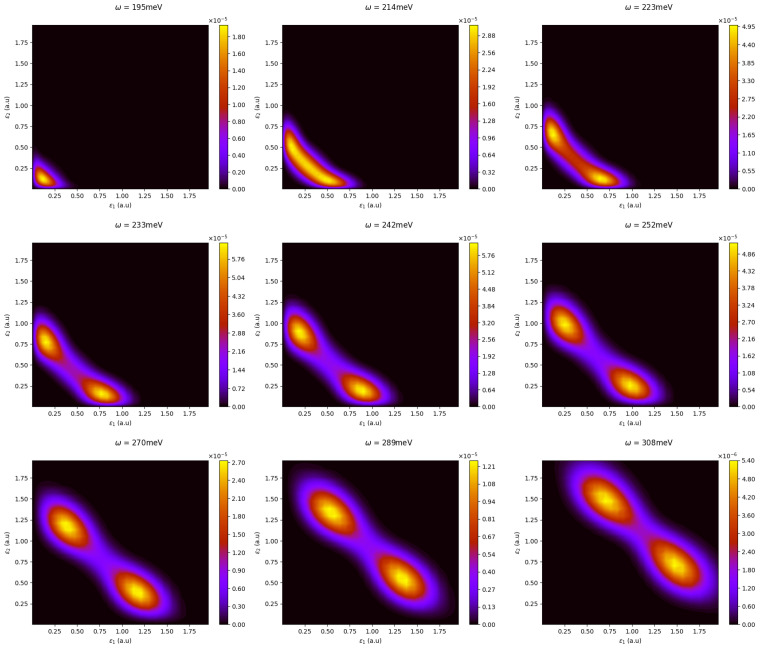
The joint photoelectron distribution patterns from the direct to the sequential DI regime as the average photon energy increases from 195 meV to 308 meV. The patterns transition from a mostly shared energy distribution to a doubly peaked distribution.

**Table 1 materials-16-01405-t001:** The s.a.u. for μ=0.1 and κ=0.5 corresponding to a CdSe QD semiconductor type.

1 s.a.u.	Conventional Units
length	2.645 nm
time	6.047 fs
energy	108.84 meV
Intensity	4.119 $\times \;10^{7}$ W/cm${}^{2}$

**Table 2 materials-16-01405-t002:** The QD physical parameters used are $V_{0}=-542.2$ meV and $r_{q}=3.2$ nm. The entry values are the energies of the states in meV.

State	QD	QD${}^{+}$
Ground	−365.2	−232.9
1st (Excited)	−230.01	−69.7
2nd (Excited)	−229.95	−3.3
3rd (Excited)	−229.84	

## Data Availability

Not applicable.

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
