# Peer review of "Dynamics of Correlated Double-Ionization of Two-Electron Quantum Dots in Laser Fields"

_materials, 2023, doi:10.3390/ma16041405_

Round 1
Reviewer 1 Report
Decision:
Minor Revision
Comments
The authors reported Dynamics of correlated double ionization of Two-Electron
Quantum Dots in laser Fields, this research article can be useful for the scientific community after some revision. The authors should address the following points outlined below to improve the scientific quality. After the suggested revisions are carefully addressed, this work may be considered for publication.
1. Figure one sketch is not clear. Its hard to read the figure . Increase the font size in inset of the figure .
2. Line 64 and 67 author mentioned section 2 and section 3 respectively. However, there is no mention of section 1 in the introduction section.
3. Which software author used for the theoretical studied? Mentioned in method section.
4. All the figures(2-5) are placed together and mentioned later in the text. Adjust the figure properly after mentioning in the text.
5. In result and discussion, line 236 the authors mentioned about identify the characteristic features of the direct and sequential DI radial patterns, represented by the 195 meV and 308 meV pulses; However, author should explain more in details why 2 peaks was achieved in 308 mev.
6. Only 16 refs were cited. Author should cite more references.
7. Lastly there are several grammatical errors. A proofread is required.
Reviewer 2 Report
In this study, the authors present computational results the dynamics of quantum dot systems with two excited electrons. The authors demonstrate by the time dependence of the correlated electrons and that as the excitation laser pulse power increases, the behavior of the excited electrons change. These results are interested, and merit publication.
There are two main criticisms of the study that I have, and addressing both should greatly improve the manuscript.
First, does the material of the quantum dot matter? There was no discussion about the material, such as cadmium selenide or indium phosphide quantum dots, but electron energies and atomic numbers are discussed. To myself, this was the biggest question while reading the manuscript. If the material does not matter, this should be explained why this general approach should be relevant for all quantum dots, or why numerical values were selected that should depend on the nature of the material. This would make the study more broadly appealing.
The second criticism that I have in a general criticism of computational manuscripts. These results, while interesting, should be verified experimentally. While it is likely beyond the scope of this manuscript to verify these results experimentally, the authors should outline what experiments should be done to help support their computational results.
Also, 16 references is a very small number of references, and I'm sure the authors could tie their results to more literature to help put their work in broader context.
Reviewer 3 Report
Reviewer’s comments on the manuscript
Dynamics of correlated double ionization of Two-Electron Quantum Dots in laser Fields
by A. Prior and L.A.A. Nikolopoulos submitted to Materials
Dear Authors and Editor!
The work under consideration is devoted to an interesting and most fundamental problem which may be very interesting to the audience. However, it appears to me that both discussion and figures should be improved to achieve better clarity. Please find below the questions that have be addressed.
1) Though the manuscript is already rather bulky, it still looks somewhat incomplete in terms of obtained results. There is no visible bridge towards possible experimental observation of the effects under discussion (if such experiments are feasible), and no details were provided regarding further theoretical analysis required to establish observable features. In my opinion, the discussion should be more detailed and explicit. Please comment on it.
2) In the manuscript, Authors only consider fixed excitation intensity of ~640 kW/cm2. Why is it so? Does it affect the balance between direct and sequential ionization of the QDs? And is it possible to express the problem in terms of photon flux (e.g. photons per QD per second) and certain relaxation times?
3) Figs. 2 & 3 are overloaded. Is it really necessary to show all 12 frames for each case? It might be better to merge existing data into a single figure so that direct and sequential ionization processes are visualized side-by-side.
4) List of references seems to be strongly polarized Are there any competing groups in the field of your research? It can hardly be unnoticed that cited references 1-5, given in the introduction, are quite general, and so are refs.14-16 in the conclusion, while all the references directly related to the discussion (Refs.6-13, except for Ref.11) are self-citations. Please comment.
Thus, I suggest that the manuscript should be published after a major revision intended to highlight exact results of the work and to bring it closer to a general reader.
Round 2
Reviewer 3 Report
Dear Authors and Editor!
Thank you very much for your time and effort! I believe the manuscript can be published in "Materials" in its present form.
Author Response
Dear Editor,
we thank the reviewer about his opinion that the manuscript should be published 'in its present form'.
sincerely,
The Authors